# Ongoing Research on the Role of Gintonin in the Management of Neurodegenerative Disorders

**DOI:** 10.3390/cells9061464

**Published:** 2020-06-15

**Authors:** Muhammad Ikram, Rahat Ullah, Amjad Khan, Myeong Ok Kim

**Affiliations:** Division of Applied Life Science (BK 21), College of Natural Sciences, Gyeongsang National University, Jinju 52828, Korea; qazafi417@gnu.ac.kr (M.I.); rahatullah1414@gnu.ac.kr (R.U.); amjadkhan@gnu.ac.kr (A.K.)

**Keywords:** gintonin, neurodegenerative disorders, lysophosphatidic acid receptors

## Abstract

Neurodegenerative disorders, namely Parkinson’s disease (PD), Huntington’s disease (HD), Alzheimer’s disease (AD), and multiple sclerosis (MS), are increasingly major health concerns due to the increasingly aged population worldwide. These conditions often share the same underlying pathological mechanisms, including elevated oxidative stress, neuroinflammation, and the aggregation of proteins. Several studies have highlighted the potential to diminish the clinical outcomes of these disorders via the administration of herbal compounds, among which gintonin, a derivative of ginseng, has shown promising results. Gintonin is a noncarbohydrate/saponin that has been characterized as a lysophosphatidic acid receptor (LPA Receptor) ligand. Gintonin may cause a significant elevation in calcium levels [Ca2+]i intracellularly, which promotes calcium-mediated cellular effects via the modulation of ion channels and cell surface receptors, regulating the inflammatory effects. Years of research have suggested that gintonin has antioxidant and anti-inflammatory effects against different models of neurodegeneration, and these effects may be employed to tackle the neurological changes. Therefore, we collected the main scientific findings and comprehensively presented them, covering preparation, absorption, and receptor-mediated functions, including effects against Alzheimer’s disease models, Parkinson’s disease models, anxiety and depression-like models, and other neurological disorders, aiming to provide some insights for the possible usage of gintonin in the management of neurodegenerative conditions.

## 1. Introduction

With an increasingly aged population, the prevalence of neurodegenerative disorders is growing proportionally. According to the World Health Organization, the population above the age of 60 will double from 12% to 22% from 2015 to 2050, reaching almost 2 billion worldwide. In the developed world, occurrences of neurodegenerative diseases have significantly increased, due to higher life expectancies. In the case of Alzheimer’s disease, it is now more than 102 years since it was first explored and described by Dr. Alzheimer (1906) [1].

Neurodegenerative disorders are categorized as a group of diseases which slowly bring about the loss of neuronal cells [2]. The mechanisms behind the onset of neurodegenerative conditions have not been fully explored; however, elevated reactive oxygen species has been suggested as one of the potential factors in neurodegeneration [3]. Normally, oxygen is required for energy metabolism and the functioning of all eukaryotic entities [4]. Along the respiratory chain, oxygen is moderately reduced at a low ratio into superoxide, which can be converted into ROS. Cell metabolism may produce another form of reagent, known as reactive nitrogen species (RNS). The ROS and RNS at normal concentrations have been suggested to facilitate several activities, including signal transduction, the induction of mutagenic response, and the induction of defense against pathogens [5]. Oxidative and the anti-oxidative mechanisms are normally balanced by certain known elements, such as nuclear factor erythroid-2 related factor-2 (Nrf2), and Heme-oxygenase-1 (HO-1). The overproduction of ROS and/or suppression of the antioxidant defense mechanism may become harmful, and is known as oxidative stress [6]. Under elevated ROS conditions, free radicals could pass through the plasma membrane, destroying the cell membrane potentials via lipid peroxidation, causing structural protein misfolding and aggregation, and may oxidize the RNA/DNA to intrude the transcription process and cause gene mutation. Collectively, the elevated oxidative and nitrosative stress may induce cellular abnormalities, impair the DNA repairing [7], and lead to dysfunction of mitochondrial energy production [8], all of which may aid in the progression of aging processes and neurodegenerative disease [9].

The roots, leaves, and stems of ginseng have been used as a medicine for the last 2000 years in Japan, Korea, and China, and are among the most accepted herbal medicines. The antidepressive, anxiolytic, memory-enhancing effects of ginseng have been fully addressed in Ben Cao Gang Mu, written by Shi-Zhen Li, which is a well-known text on herbal medicine, published during the Ming dynasty in China. The *Panax ginseng* is cultivated in Korea, China (northeast), and the coastal region of Siberia. The *Panax quinquefolium* (American ginseng) is cultivated in Canadian and American coastal areas, and *Panax notoginseng* is cultivated in China. Panax means “cure-all” in Greek. Ginseng has been shown to produce different effects, such as antioxidant, antiaging, immunomodulatory, anti-inflammatory, vasodilatory, adaptogenic, anticancer, antifatigue, and antidepressive, in experiments on rodents [10]. The main ingredients of *P. ginseng* are acidic polysaccharide and ginsenoside. The effects of its ingredients have been elucidated in different disease models, but more efforts are needed to explore the underlying mechanisms responsible for these effects. Recently, an ingredient from *P. ginseng* was successfully isolated, called gintonin, which is a noncarbohydrate/nonsaponin polymer that was found to be a ligand of G protein-coupled LPARs [11]. Gintonin has been shown to elicit a robust elevation in the intracellular calcium level [Ca2+]i, which may further induce calcium-dependent cellular processes via cell surface receptors and ion channels, thereby inducing anti-inflammatory activities, mainly by preventing the mitogen-activated protein kinases (MAPKs) and nuclear factor-kappa B (NF-κB) signaling in lipopolysaccharide-induced RAW 264.7 cells. Similarly, it has been suggested that gintonin triggers the release of different neurotransmitters such as dopamine, catecholamine, and gliotransmitter in primary astrocytes and PC12 cells; after intraperitoneal (i.p.) injections of gintonin, mice showed markedly elevated dopamine levels in serum [12]. In this comprehensive review, we have summarized all the current research works conducted on gintonin in the management of neurodegenerative diseases. Regarding our literature search, the inclusion and exclusion criteria are given in the methodology section. All papers have been summarized, covering the effects of gintonin against neurodegenerative disorders. A simple illustration has been given, showing the process of neuronal cell loss in neurodegenerative conditions (Figure 1).

## 2. Methods

### 2.1. Search Strategy

We extensively searched for and reviewed articles discussing the protective activities of gintonin in animal models. The papers were collected from different independent databases, e.g., PubMed (https://pubmed.ncbi.nlm.nih.gov), Google Scholar (https://scholar.google.co.kr), and Web of Science (https://apps.webofknowledge.com). We used the keywords “Neuroprotection”, “Neuroprotective”, and “gintonin” in our literature search. One reviewer (Muhammad Ikram) analyzed the animal and cellular studies by screening the abstracts of the searched articles.

### 2.2. Inclusion and Exclusion Criteria

a. Laboratory rodents of any species and age, used as neurodegeneration models, were covered. b. Comparisons among the control, toxin-treated, and toxin + gintonin cotreated groups were included. The control group was injected with a physiological saline/placebo or a similar vehicle. The toxin-injected group was used as a model group. Drug administration, route of administration, and duration of the dose were not considered. Only experimental studies which highlighted the effects of gintonin on animal models of neurodegeneration were included. No review articles, duplicate references, or studies with incomplete and incorrect data were included.

## 3. Isolation of Gintonin

Gintonin was first isolated by Pyo, Mi-Kyung [13]; red ginseng roots were ground (>3 mm) and refluxed with 80% methanol (MeOH) three times for almost 8 h at 80 °C. The methanolic extract (6.2 kg) was concentrated in vacuo and separated into water and n-butanol (n-BuOH, 908 g). After concentration, the extract was run in a silica gel column and eluted with chloroform (CHCl_3_): MeOH: water (H_2_O) = 13:7:2. The effect of each isolate was analyzed on endogenous Ca^2+^-triggered Cl^−^ channel (CaCC) in oocytes (*Xenopus* sp.). The fraction which activated the CaCC was further fragmented via a column and eluted with ethyl acetate (EtOAc): ethanol (EtOH):H_2_O = 1:3:0.5. To filter the mini parts, the isolate which activated the CaCC was dialyzed at 4 °C for 8 h with a 1000-fold excess of distilled water using a spectra/pore dialysis membrane (molecular weight cut off 6000–8000) (Spectrum Laboratories Inc., Rancho Dominguez, CA, USA). The remaining material was named “crude gintonin”, which was further refined into pure gintonin through a series of processes (Figure 2).

## 4. Absorption of Gintonin

For almost all systemic treatments, the choice of route for drug administration is oral administration, because of its advantages and safety compared to other routes; herbal medicinal preparations are no exception, since the majority are decocted with water to extract the active ingredient(s) and for ease of usage. Orally administered medicines are normally absorbed by the gastrointestinal layer. Lee et al. showed that gintonin is significantly absorbed by the intestine [14].

## 5. Lysophosphatidic Acid Receptor, Gintonin, and Neurological Disorders

For the synthesis of phosphatidic acid, lysophosphatidic acid (LPA) acts as a precursor, which is widely present in the brains of mammals. LPA, which is a phospholipid, yields different types of effects in the peripheral and central nervous system [15]. These functions are accomplished via LPA receptors [16]. The prominent function of the LPA receptor is to elicit intracellular calcium [Ca^2+^]i) transients via different mechanisms. The intracellular calcium transient is combined with different cellular effects in the growth of the brain. Moreover, LPA receptors play a prominent role in the neurogenic processes of the hippocampal region, cognitive, and memory processes. It has been suggested that LPA1 receptor-deficient mice showed reduced neurogenesis at the hippocampal level [17] and impaired spatial working memory and cognitive dysfunction [18,19]. It has been suggested that gintonin may potentiate the LPA receptor with a significantly higher affinity. The methanolic extract of gintonin caused LPAs to be moved from the protein compartment of gintonin. The LPA complexes making up gintonin have different affinities for LPA receptors, and the various proteins in gintonin triggered activations. It was hypothesized that the coexistence of LPAs with different proteins in ginseng may be valuable for the development of new chemical entities targeting LPA receptors [20] (Figure 3).

## 6. Effects of Gintonin in Animal Models of Parkinson’s Disease

Parkinson’s disease is categorized by the degeneration of the dopaminergic neurons and the accumulation of a-synuclein into Lewy bodies and neuritis in various brain areas, such as the substantia nigra (SNpc) of the midbrain, and reduced dopamine levels in the striatum [12]. The most well-known and prominent symptom of PD is impaired movement, such as muscle stiffness, involuntary tremors, and postural instability. Apart from motor dysfunctions, some nonmotor symptoms may also appear, including executive dysfunction, sluggishness of cognition, genitourinary problems, and emotional disturbances. Multiple factors may be involved in the progression of PD, including mitochondrial dysfunction, glutamate toxicity, apoptotic cell death, elevated oxidant stress, proteasomal dysfunction, and environmental factors [21].

PD is the second most common neurodegenerative disease after AD; many studies have been conducted to explore the basic pathology of the disease. One study targeted the lysophosphatidic acid receptor agonist in a 1-methyl-4-phenyl-1,2,3,6-tetrahydropyridine (MPTP) injected mouse model of PD. According to the findings of the authors, gintonin significantly restored the expression of tyrosine hydroxylase (TH), upregulated the expression of Nrf2, and downregulated the elevated levels of MAP kinases. Phospho- C-Jun *N*-terminal kinase (p-JNK), for example, reduced the expression of activated microglial cells and regulated the motor function of the mice in the experiment [21]. Similarly, recently, we conducted a comprehensive study of gintonin in an MPTP-injected, PD mouse model. We showed that gintonin significantly improved locomotory function, improved TH level, reduced elevated oxidative stress, inhibited inflammatory mediators, and overall conferred neuroprotection to the mouse brain against MPTP-induced dopaminergic neurodegeneration [3] (Figure 4).

## 7. Effects of Gintonin Against Alzheimer Disease Pathology

Alzheimer’s disease (AD), for which the prime clinical symptoms include memory dysfunction and progressive behavioral alterations, is one of the most common neurodegenerative disorders [22]. Since the last century, AD has become a threat to the health of the elderly worldwide. AD has two main features, i.e., extracellular senile plaque caused by the deposition of amyloid β-peptide (Aβ), and intracellular neurofibrillary tangles (NFTs) formation composed of hyperphosphorylated tau [23]. Aβ may induce toxicity in the central nervous system, which eventually leads to neuronal cell loss; the neurotoxicity may result in the hyperphosphorylation of tau and the subsequent accumulation and establishment of NFTs, which affects the structure of the microtubules of the neuronal system, and disturbs the transportation of axons, causing neurodegeneration and memory dysfunction [24]. Several studies have sought to find a therapeutic strategy against AD, but there is presently no effective strategy against the disease [25]. Multiple factors are involved in brain development and the processing of information, one of which is acetylcholine, which is openly found in the brain, and which plays a prominent role in cerebral development, cortical function, regulation of cerebral blood flow, and regulation of overall cognitive and memory performance. It also regulates the functional and structural remodeling of cortical circuits by establishing synaptic contacts in networks of cells [26]. In AD cases, it has been suggested that the level of brain acetylcholine is significantly reduced, affecting cognitive and memory processing; the reduction in acetylcholine occurs mainly because of dysfunction of the cholinergic system [27]. Recently, a study was conducted on the role of gintonin against cholinergic dysfunction in AD cases, both in animal and cellular models of AD. According to the findings, the oral administration of gintonin triggered [Ca^2+^]i transient in mouse hippocampal neural progenitor cells (NPCs), which subsequently regulated the release of acetylcholine through LPA receptors. Gintonin significantly regulated scopolamine-prompted memory dysfunction and modulated amyloid-β (Aβ)-induced cholinergic impairment, reduced the acetylcholine concentration, decreased choline acetyltransferase (ChAT) function, and upregulated the acetylcholine esterase (AChE) in mouse brain. The same phenomenon was evaluated in a transgenic AD mouse model; it was shown that gintonin regulated cholinergic impairment [28]. Collectively, these findings suggest that the activation of the LPA receptor by gintonin is responsible for the regulation of cholinergic functions. The effects of gintonin on cholinergic functions in an animal model of AD are depicted in Figure 5.

## 8. Effects of Gintonin Against Huntington’s Disease

Huntington’s disease (HD) is a congenital condition which is characterized by involuntary movements, cognitive dysfunction, and psychiatric illness. It is instigated by an irregular expansion of CAG (glutamine) trinucleotide repeats in exon 1 of the huntingtin gene at the 4p16.9 location [29]. The aggregation of the mutant huntingtin-protein leads to different pathogenic effects, including deadly neuronal accumulation, transcriptional dysfunction, mitochondrial malfunction, metabolic dysfunction, altered axonal transport, and synaptic dysfunction in the striatal and cortical region of the brain [30]. Tetrabenazine is the only US Food and Drug Administration-approved drug for the management of HD, while antipsychotic drugs including aripiprazole and olanzapine are thought to be possible candidate drugs for the management of psychotic disorders. However, there is a risk of serious side-effects including dizziness, depression, fatigue, or Parkinson’s like symptoms, with the administration of Tetrabenazine. To analyze the effects of gintonin against HD, Jang et al. conducted a comprehensive study, showing that gintonin has significant protective effects with a wide range of therapeutic potential in 3-nitropropionic acid (3-NPA)-induced striatal toxicity by reducing oxidative stress and neuroinflammation, possibly by upregulating the level of LPA. Furthermore, gintonin showed protective effects in STHdh cells and in an adeno-associated viral (AAV) vector-infected model of HD [31]. Thus, gintonin may be a novel therapeutic drug to treat HD-like symptoms.

## 9. Effects of Gintonin against Multiple Sclerosis

Multiple sclerosis (MS) is a demyelinating and inflammatory disorder of the CNS, affecting a large number of individuals globally [32,33]. There have been many hypotheses regarding the pathogenesis of the disease. It has been suggested that the penetration of immune cells, including autoreactive T cells and interleukin (IL)-17-producing Th17 subsets, into the central nervous system (CNS) via the blood-brain barrier is the main factor involved in the pathogenesis of MS. The triggered immune cells can cause the secretion of the inflammatory reagents, which may induce inflammatory and demyelinating effects in the cerebrospinal region [34]. To study the possible effects of gintonin, mice was administered 100 mg/kg, orally, for 10 days before immunizing with myelin basic protein (MBP) 68–82 peptide. According to their findings, gintonin provided protection to the spinal cord by inhibiting the activation of microglial cells, decreasing the concentration of mRNA expression of the inflammatory mediators, such as interferon-γ, IL-6, and cyclooxygenase-2, and inducing the expression of protective mediators, including insulin-like growth factor-1, and vascular endothelial growth factor-1. The study suggested that there was a significant inhibition in the levels of p38 mitogen-activated protein kinase and nuclear factor-kB signaling genes in the immune cells. Collectively, it was suggested that gintonin improves the demyelination in autoimmune encephalomyelitis via the inhibition of MAPK and NF-kB, suggesting gintonin as a therapeutic candidate for the management of autoimmune disorders, although further studies should be conducted to explore its mechanisms.

## 10. Effects of Gintonin on Anxiety and Depressive Behavior in Mice

Depression is one of the leading causes of disability worldwide, with a prevalence of 14.6% in developed countries [35]. Although significant efforts have been made to lighten the load of the disease symptoms, only one-third of patients achieve remission after an adequate trial period with first-line antidepressant treatment [36]. So, a novel, effective, and safe drug with antidepressant activity is crucially needed [37]. Many mechanisms have been explored regarding the pathophysiology of depression; some studies have suggested that the activation of immunity and neuroinflammation may be associated with the development of the pathology of depression [38]. The current meta-analysis data show that compared with normal individuals, the levels of various inflammatory mediators (including tumor necrosis factor (TNF)-α, IL-6, and IL-1 receptor antagonists) in depression patients are increased [39]. Moreover, there was a higher inflammation in half of resistant patients observed in a study [40]. On the other hand, most antidepressants can regulate peripheral inflammatory conditions, but remains to be determined whether these anti-inflammatory effects induce the expected antidepressant effects [41]. Peripheral inflammation may contribute to the emergence of depressive-like conditions via different mechanisms, such as kynurenine signaling [42], changes in neuronal plasticity [43], activation of microglia [44], and the endocrine system. It has been suggested that 95% of the serotonin in the body is produced in the gut, and then stored in gastrointestinal enterochromaffin (EC) cells [45]. The activation of the gastric EC cells in the gastrointestinal tract (GIT) causes the release of serotonin (5-HT) via the calcium-reliant mechanism [46] The 5-HT secreted from intestinal EC cells promotes intestinal motility and the release of enzymes and juices in the digestive system [47]. Currently, the effects of 5-HT released from the gastric systems on the CNS have not been fully explored, but most recent findings have suggested that intestinal 5-HT affects the vomiting center in the brain via a gastric afferent mechanism to the CNS [48]. Moreover, it has been suggested that LPA1 receptor-null mice drink a higher amount of alcohol during alcohol withdrawal tests, and show depressive effects [49]. Also, the concentration of 5-HT in the plasma was substantially reduced in patients with depression [50]. The association between 5-HT from gastric EC cells and alcohol withdrawal-induced anxiety-like effects has not yet been fully elucidated. BON cells treated with gintonin increased the intracellular calcium level and the release of 5-HT in a dose- and time-dependent manner via the activation of the LPA receptor. The administration of the gintonin-enriched fraction (GEF) increased plasma serotonin, and decreased immobility times in tests used for the evaluation of depression during alcohol withdrawal. The main idea was that the gintonin-induced release of catecholamine, acetylcholine and glutamate in the brain may alleviate depression-like effects. The conclusion was that the regulation of depression induced by GEF-mediated alcohol withdrawal may be regulated by the release of 5-HT from EC cells. It is therefore possible that GEF of ginseng can regulate depression (Figure 6).

## 11. Effects of Gintonin against Hypoxic and Re-Oxygenation Stress via the Activation of Astrocytic Glycogenolysis

Cerebral hypoxia and reoxygenation (H/R) injury are the causes of several disorders, including high altitude cerebral edema, traumatic injuries, mountain sickness, respiratory disorders, cardiac arrest, obstructive sleep apnea, and ischemic stroke [51]. They are related to neuronal cell loss and proteolytic cascade, characterized by the release of cytochrome c, caspase-3, and DNA disintegration [52]. As neuroinflammation and apoptosis become more severe, brain tissue and cells are affected. Similarly, in ischemia-reperfusion injury, the main cause of neuronal injury is not the ischemia itself; the destruction occurs after the recovery of the blood supply, possibly due to elevated reactive oxygen species (ROS) [53]. Astrocytic cells are the most abundant cells in the CNS; they exhibit a variety of functions, both in normal and abnormal situations [54]. The main characteristics of astrocytes are its ability to reserve glycogen for the supply of energy to the brain and to support neurons and cognitive functions [55] in pathological situations such as hypoxic ischemia [56]. According to the astrocyte-neuron lactate shuttle hypothesis, astrocytes convert glycogens to lactate, which is secreted into neuronal cells for metabolic processes [57]. So, the principal function of brain astrocytic glycogen is as an energy reservoir for different conditions in which glucose is required, such as hypoxic ischemia and hypoglycemia. Studies have suggested that brain injury can elevate the expression of LPA receptors in mice cortex and spinal cord astrocytes. The LPA receptors found in astrocytes may play vital roles in the metabolic processes of the brain under normal and pathological conditions. However, little is known about LPA-mediated astrocytic glycogenolysis in disease and normal conditions. Since astrocytes express LPA receptors [58], and gintonin acts as a ligand for LPA receptors [59], it follows that gintonin mediated-[Ca^2+^]i transients may help in the regulation of the metabolic processes of astrocytic cells. This hypothesis was confirmed in a study that analyzed whether gintonin mediates Ca^2+^-dependent glycogenolysis in the astrocytes. The authors analyzed the effects of gintonin on astrocytic glycogenolysis protecting the astrocytes against hypoxia injuries. They have found that gintonin promotes astrocytic glycogenolysis via LPA receptor-mediated Ca^2+^-signaling, thereby modulating the key enzymes involved in glycogenolysis. Gintonin-mediated astrocytic glycogenolysis stimulates adenosine triphosphate (ATP) production and glutamate uptake, and enhances cell viability under hypoxic conditions. The production of ATP and the uptake of glutamate under hypoxic conditions confers protection against hypoxic injury upon astrocytes. It has therefore been suggested that gintonin-mediated [Ca^2+^]i transients regulated by LPA receptors may protect the astrocytes via LPA receptors under hypoxia conditions (Figure 7).

## 12. Effects of Gintonin against Synaptic Dysfunctions in Neurodegenerative Conditions

The basic units of cognitive performance and information sharing in the brain are called synapses, which are made of post- and pre- synaptic parts. Various proteins have been implicated in the processing of the synapses, such as SNAP-25 (synaptosomal-associated protein of 25), PSD-95 (postsynaptic density protein 95), synapsin 1 and chromogranin B (synaptic vesicle proteins), all of which are reduced in AD brain [60,61]. Previous findings have suggested that the cortex and hippocampus of AD brains are drastically affected, i.e., in the form of synaptic dysfunction, while the occipital cortical region is the least affected area [62,63]. Furthermore, an experimental AD model presents impaired long-term potentiation (LTP) and long-term depression (LTD), and reduced synaptic functions [64]. These findings suggest that synaptic dysfunction occurs in AD at the early stages of disease progression. However, neuronal cell death is not enough to show robust synaptic dysfunction, indicating that synapses are gradually removed before cell death [64]. It has been suggested that some living neurons lose synapses in neurodegenerative conditions [65]. Without causing neuronal cell loss, the accumulation of Aβ and hyperphosphorylation of Tau can trigger synaptic loss [66,67]. Studies have suggested that LPA receptor activation plays a pivotal role in the early stages, including synaptogenesis, and the morphologies of hippocampal and cortical neurons [68]. LPA receptor-1-deficient mice presented marked dysregulations in their behavioral functions, as shown in prepulse inhibition [69], hippocampus-mediated spatial memory [70], and pain sensation tests [71]. It has been suggested that in LPA receptor-deficient mice, there is reduced neurogenesis in the embryonic and adult cerebral cortex and hippocampus [17]. Another study indicated that LPA receptor-1 deficient mice presented dysregulated fear extinction that was worsened with the administration of LPA receptor-1 antagonist, suggesting that LPA receptor-1 plays a critical role in the expression of emotional [72]. The effects of LPA receptor activation in the synaptogenic processes of central synapses after synapse maturation have not been explored, but these effects may be due to the level of LPA receptors in the hippocampus being reduced after growth. Moreover, no agonist of LPA receptors with a good affinity in the hippocampus has been proposed. In one study, the authors showed that gintonin activated the LPA receptor, thereby regulating the hippocampal synapses [73]. They suggested that the activation of LPA receptors by gintonin markedly elevated inhibitory and excitatory neurotransmission in the central synapses through different mechanisms. The gintonin-activated LPA receptor showed synaptic improvement and increased neuronal excitability in a phospholipase C-dependent mechanism. The authors concluded that gintonin acts as a LPA receptors agonist, enhancing synaptic transmission in the central synapses, which may serve as a major entity to regulate the synaptic effects under pathological conditions, including neurodegenerative diseases [73].

## 13. Effects of Gintonin against Different Models of Neurodegeneration

Inflammation is the main characteristic of Alzheimer’s disease, Parkinson’s disease, and traumatic brain injury [74]. Studies have suggested that multiple factors are involved in the pathophysiology of neurodegenerative disorders, such as activated microglia; furthermore, astrocytosis (gliosis) play a prominent function in the onset of neuroinflammation, which may induce the release of different inflammatory cytokines, ultimately leading to neurodegeneration and cognitive dysfunction [2]. To show its possible effects against neurodegenerative diseases, gintonin has been used against established models of neurodegeneration. Nam et al. analyzed the effects of gintonin against d-galactose-induced neurodegeneration; they showed that gintonin significantly improved the hippocampal LTP, and improved neurogenesis and cognitive performance in mice [75]. Similarly, in another study, it was suggested that gintonin reduced scopolamine-induced memory dysfunction in mice. The authors indicated that gintonin markedly regulated amyloid-β-induced cholinergic impairment, e.g., through reduced acetylcholine level and diminished ChAT activity, and increased the level of AChE [28]. Moreover, it was shown that gintonin regulates the expression of VEGF in astrocytes, which might provide protection against hypoxic injury [76]. It has been suggested that gintonin is able to protect mouse brain against methylmercury-induced neurotoxicity and learning processes [77]. One of the most common mechanisms, which has been extensively highlighted in a study of gintonin, is its antioxidant effects; a simple diagram is given here (Figure 8). The diagram suggests that gintonin markedly upregulates the expression of Nrf2/HO-1, thereby enhancing the expression of some major enzymes, e.g., glutamate cysteine ligase (GCL), superoxide dismutase 1 (SOD1), glutathione (GSH) and glutathione peroxidase (GPx).

## 14. Conclusions and Future Considerations

The ability of gintonin to modulate multiple aspects of neurodegenerative conditions, including antioxidant mechanisms, calcium regulation, antineuroinflammation, and the regulation of survival and apoptotic mechanisms, makes it suitable for the treatment of neurodegenerative conditions. Moreover, it has been suggested that it can potentiate LTP and regulate hippocampal neurogenesis. Its safety, efficacy, low cost and wide availability make it a candidate drug for the management of neurodegenerative disorders which merits further study, including in preclinical and clinical studies.

Several things must be considered to elucidate the efficacy and safety of gintonin for clinical use. The majority of the reports with gintonin studies were conducted either in preclinical disease models or in cellular models. Here, animal or cellular models of disease were pre-/post- treated with gintonin, and the onset of disease was evaluated. Only one study, with many shortcomings, described the beneficial effects of gintonin in humans [78]. The potential neuroprotective effects of gintonin need further elucidation in terms of clinical aspects.

More work is needed to fully address the question of whether gintonin is protective or restorative; several studies have shown that gintonin can be cotreated with toxic compounds. More comprehensive pharmacokinetic and pharmacodynamics studies should be conducted to better elucidate its mechanisms of action and absorption. More effort is needed to develop new methods to prepare pure gintonin on an industrial scale, especially for long-term in vivo and human studies, due to the complex nature of the purification processes. Efforts may be made to synthesize gintonin chemically, which will save costs and facilitate the large-scale production of the compound. The isolation and purification of gintonin from plants has advanced rapidly in recent years, but current protocols need more accuracy and precision.

Moreover, the current use of gintonin against neurodegenerative disorders is quite generalized, and a deeper understanding of its mechanisms is needed. The current findings indicate its beneficial effects on AD, but the exact mechanisms are still not clear. For example, does it affect the enzymes responsible for the production of Aβ? If so, which specific enzyme is involved in its mechanism of action? These and related questions remain unanswered. Similarly, it has been suggested that gintonin may protect dopaminergic neurons, but the mechanism remains unclear. Before gintonin becomes a protective compound, these questions must be investigated. If such deep study is successful, it might be a miracle. In most studies, gintonin was taken orally. Although gintonin has a significant protective effect in different neurodegenerative models, its bioavailability has not yet been analyzed. If its bioavailability is not significant, then how could it be improved? Notably, the encapsulation of gintonin in poly(lactide-coglycolide) (PLGA) or gold nanoparticles conjugated to polyethylene glycol (PEG) has not been presented to date. A lot of research is needed to explore parameters related to safety, bioavailability, and efficacy. Collectively, the current findings suggest that gintonin is an effective drug against different kinds of neurodegenerative conditions, signifying the role of gintonin in the management of neurodegenerative conditions.

## Figures and Tables

**Figure 1 cells-09-01464-f001:**
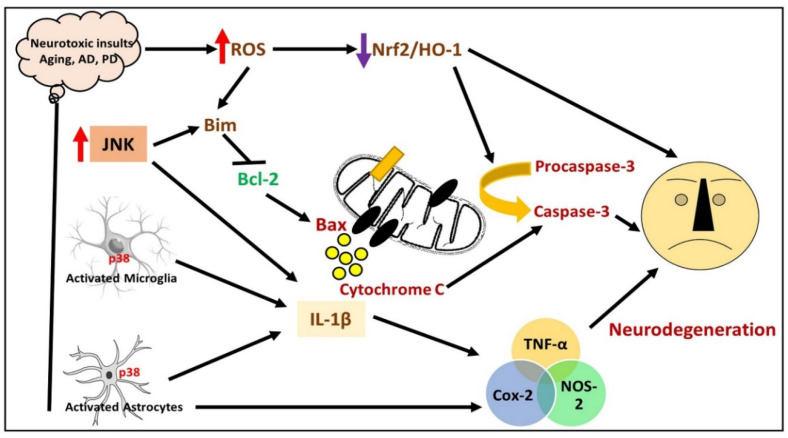
Neuronal cell loss in neurodegenerative conditions. Neurodegeneration can be induced by several stimuli; the activation of microglia and astrocytes results in the release of inflammatory mediator progression of apoptotic degeneration. Elevation in oxidative stress and the downregulation of Nrf2/HO-1 may also contribute to the process of aging. IL-1β: Interleukin-1, AD: Alzheimer’s disease, PD: Parkinson’s disease, ROS: Reactive oxygen species, Nrf2: Nuclear factor-2 related factor-2, HO-1, Heme oxygenase-1. TNF-α: Tissue necrosis factor-alpha, Cox-2: Cyclo-oxygenase-2, NOS-2: Nitric oxide synthase-2.

**Figure 2 cells-09-01464-f002:**
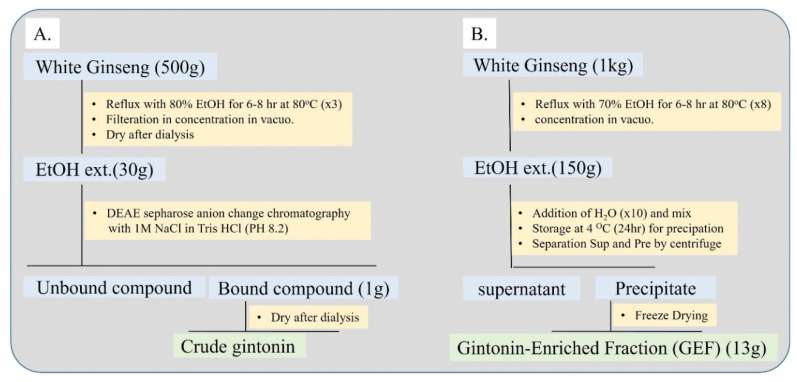
Methods for the preparation of gintonin-enriched fraction (GEF) from ginseng. (**A**) A method for gintonin fraction preparation using ethanol extraction and diethylaminoethyl (DEAE) anion exchange chromatography. Gintonin fraction was prepared from eluate of anion exchange chromatography. (**B**) A simple method for GEF using only ethanol and water. Ethanol extract is dissolved in water, leading to the formation of a precipitate, and centrifuge is used to separate the supernatant and precipitate. The precipitate from water fractionation is designated GEF with a yield of 1.3%.

**Figure 3 cells-09-01464-f003:**
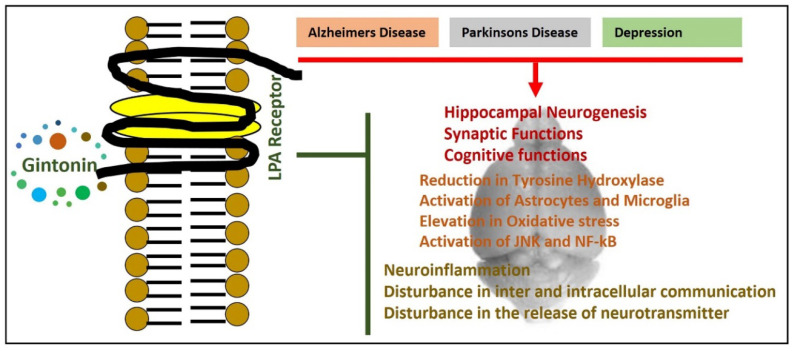
A simple diagram showing the role of lysophosphatidic acid in neurodegenerative disorders, and the possible therapeutic effects of gintonin. Red arrows show the pathological conditions, and green lines show the reversing effects of gintonin (as an LPA receptor agonist). LPA receptor: Lysophosphatidic acid receptors.

**Figure 4 cells-09-01464-f004:**
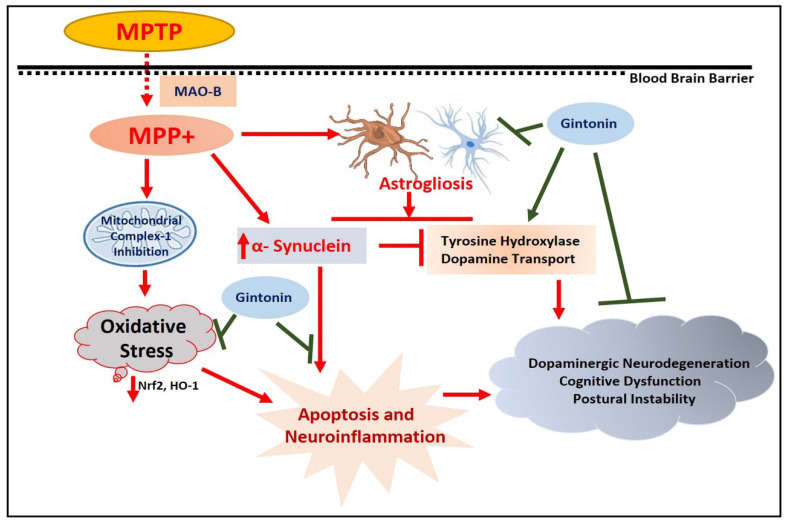
Effects of gintonin against Parkinson’s disease. A simple illustration, showing the rescuing effects of gintonin against MPTP-induced dopaminergic neurodegeneration and motor dysfunctions, covering the antioxidant, anti-inflammatory, and anti-apoptotic effects of gintonin. MPTP: 1-methyl-4-phenyl- 1,2,3,6-tetrahydropyridine), Nrf2: nuclear factor erythroid 2-related factor 2, HO-1: Heme oxygenase-1, BAX: (BCL2 Associated X, Apoptosis Regulator, Bcl-2: B-cell lymphoma 2, MPP+: 1-methyl-4-phenylpyridinium.

**Figure 5 cells-09-01464-f005:**
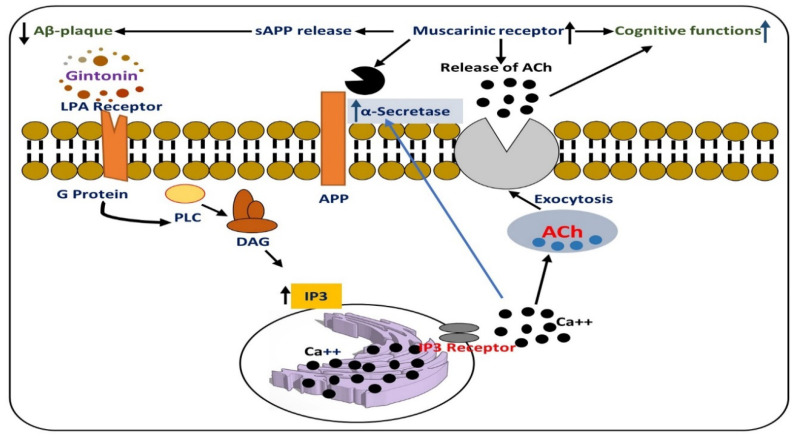
Graphical representation of the effects of gintonin against cholinergic dysfunction in AD models, possibly by regulating the LPA receptor. It has been suggested that gintonin mediates cholinergic systems by acetylcholine release and the increase of ChAT expression and protective effects against Aβ-induced and transgenic AD animal models. Gintonin-mediated activation of LPA receptors may confer neuroprotection by regulating the amyloidogenic pathway and cholinergic dysfunction in the brain. DAG: diacylglycerol, IP3: Inositol trisphosphate, LPA receptor: Lysophosphatidic acid receptors, PLC: Phospholipase C.

**Figure 6 cells-09-01464-f006:**
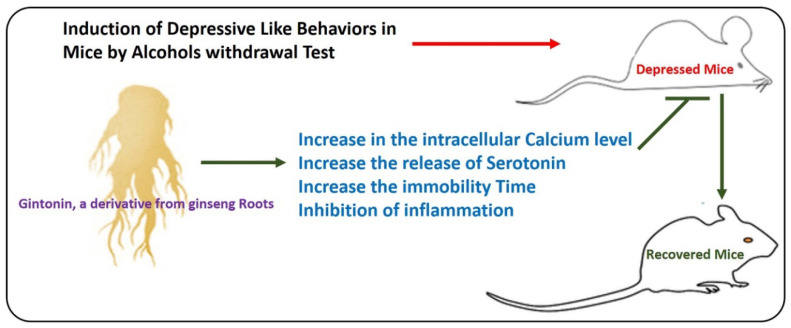
Antidepressant effects of gintonin. A simple illustration showing the protective effects of gintonin against depressive behavior in mice by increasing the intracellular calcium level, increasing the release of serotonin, and increasing the immobility time. Neuroinflammation is the main inducer of depression, but it has not been analyzed in the current studies. Gintonin may counteract depressive symptoms in mice, possibly by regulating the inflammatory mediators, increasing the immobility time, and increasing the release of serotonin and intracellular calcium levels.

**Figure 7 cells-09-01464-f007:**
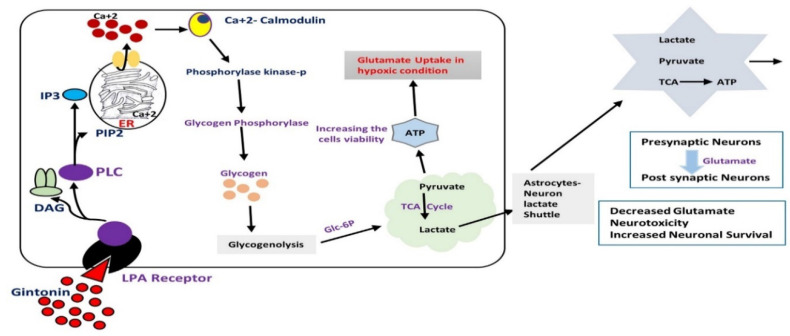
Diagram highlighting the effects of gintonin on astrocytic glycogenolysis in primary cortical astrocytes. Gintonin-mediated astrocytic glycogenolysis might be coupled with intracellular and intercellular mechanisms. Intracellularly, gintonin may reduce ATP, upregulate the glutamate level, and increase cell viability under hypoxia- and reoxygenation-induced conditions. The intercellular activities of gintonin-mediated astrocytic glycogenolysis are also tied to the astrocyte-neuron lactate shuttle, permitting neurons to use lactate under hypoxia or reoxygenation conditions and regulate glutamate toxicity by handling the unnecessary extracellular glutamate under hypoxic conditions. DAG: diacylglycerol, IP3: Inositol trisphosphate, LPA receptor: Lysophosphatidic acid receptors, PLC: Phospholipase C. PIP2: Phosphatidylinositol Biphosphate.

**Figure 8 cells-09-01464-f008:**
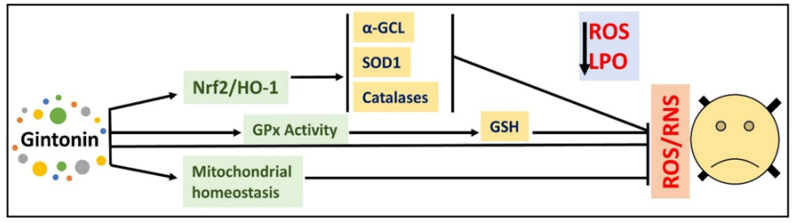
Graphical illustration showing the antioxidant potential of gintonin against elevated oxidative stress and neurodegeneration. Nrf2: nuclear factor erythroid 2-related factor 2, HO-1: Heme oxygenase-1, GCL: Glutamate Cysteine Ligase, SOD1: Superoxide Dismutase 1, RSO: Reactive Oxygen Species, LPO: Lipid peroxidation, RNS: Reactive nitrosative species, GSH: Glutathione, GPx: Glutathione peroxidase.

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
