# Peer review of "Ongoing Research on the Role of Gintonin in the Management of Neurodegenerative Disorders"

_cells, 2020, doi:10.3390/cells9061464_

Round 1
Reviewer 1 Report
The manuscript # cells-822020 entitled “Ongoing Research on the Role of Gintonin in the Management of Neurodegenerative Disorders: Focus on Future Perspectives” written by Muhammad I. et al. provides a review of currant research on gintonin, a derivative of ginseng, as possibly anti-neurodegenerative compound. In this paper authors analyze literature data concerning effects of gintonin against animal models of Parkinson’s disease, against Alzheimer disease, Huntington disease, multiple sclerosis, depression, hypoxic and re-oxygenation stress. Generally the manuscript has been written in an interesting way and provide currant stage of knowledge in the area.
Nevertheless, chapter 14.” Future considerations” in opinion of the referee is to general and very poor. It does not show any specific direction for future research. In opinion of the referee this chapter should be removed and substituted with short conclusion.
There are also some lingual problems eg.
- Line 37- it known/ it is known
- Line 41- have not yet fully explored/have not been fully explored
- line 409- some major hormones/some major enzymes
Thus the text needs lingual overview by qualified translator.
The title of the manuscript should be shortened. Since the paper poorly refers to the future perspectives for gintonin as a drug and last chapter should be removed the phrase “Focus on future perspectives” should be removed from the title.
In opinion of the referee the manuscript may be published following minor revision.
Author Response
Reviewer#1
General Response: Thank you for reviewing our manuscript. The main points have been addressed accordingly, and have been presented comprehensively. We did our best to address all your comments and present our paper in a more clear and understandable manner. All the changes have been highlighted (Blue color) and respective line numbers have been given here in the response section. Please follow our provided Line numbers, as the previous Line numbers may have changed with the suggested correction/changes. We hope that this time the paper will be appropriate for publication in the journal of “cells”.
Reviewer#1
The manuscript # cells-822020 entitled “Ongoing Research on the Role of Gintonin in the Management of Neurodegenerative Disorders” written by Muhammad I. et al. provides a review of currant research on gintonin, a derivative of ginseng, as possibly anti-neurodegenerative compound. In this paper authors analyze literature data concerning effects of gintonin against animal models of Parkinson’s disease, against Alzheimer disease, Huntington disease, multiple sclerosis, depression, hypoxic and re-oxygenation stress.
Comment#1. Generally the manuscript has been written in an interesting way and provide currant stage of knowledge in the area. Nevertheless, chapter 14.” Future considerations” in opinion of the referee is to general and very poor. It does not show any specific direction for future research. In opinion of the referee this chapter should be removed and substituted with short conclusion.
Response. The section 14 has been merged with the conclusion part, which may be noted at the section#14. Line number 408.
Comment#2. There are also some lingual problems e.g. Line 37- it known/ it is known.
Response. The point has been corrected. Line number 37.
Comment#3. Line 41- have not yet fully explored/have not been fully explored.
Response. The sentence has been revised completely. Line number 37.
Comment#4. Line 409- some major hormones/some major enzymes.
Response. The major hormones have been replaced with some major enzymes, as suggested. Line number 401.
Comment#5. Thus the text needs a lingual overview by qualified translator. The title of the manuscript should be shortened. Since the paper poorly refers to the future perspectives for gintonin as a drug and last chapter should be removed the phrase “Focus on future perspectives” should be removed from the title. In the opinion of the referee the manuscript may be published following minor revision.
Response. Thank you for your valuable comments regarding the title and the chapter#14. The Chapter 14 has been merged with the 15, and the title has been shortened, as suggested. Line number 408.
Thank you so much for your valuable and constructive comments.

Reviewer 2 Report
Gintonin has antioxidant and anti-inflammatory effects against different models of neurodegeneration, and these effects may be employed to tackle the neurological changes. Therefore, the authors collected the main scientific findings and comprehensively presented them, covering its preparation, absorption, receptor-mediated functions, including effects against Alzheimer's disease models, Parkinson’s disease models, anxiety and depression-like models, and other neurological disorders, aiming to provide some insights for the possible usage of Gintonin in the management of neurodegenerative conditions.
Major points:
- In text, all “Gintonin” and “Glycogenolysiss” hould be presented as “gintonin” and “glycogenolysiss”, respectively. “LPAR” and “LPAR1” should be “LPA receptor” and “LPA1 receptor”, respectively.
- Line 274-276: “As, a current meta-analysis data has suggested that in depression, there is an elevation in the levels of different inflammatory-mediators including, tumor necrosis factor TNF-α, interleukin IL-6, and the IL-1 receptor antagonist compared to normal individuals” should be changed into “Therefore, the current meta-analysis data shows that compared with normal individuals, the levels of various inflammatory mediators (including tumor necrosis factor (TNF)-α, IL-6 and IL-1 receptor antagonists) in depression patients have increased.
- Line 296-298: “The main concern over here is that, the possibility that gintonin-triggered release of catecholamine plasma, and acetylcholine and glutamate in the brains, may alleviate the depressive-like effects.” should be changed into “The main concern is that the gintonin-induced release of catecholamine, acetylcholine and glutamate in the brain may alleviate depression-like effects.”.
- Line 298-300: “Conclusively, the GEF-mediated regulation of depression triggered by alcohol withdrawal may be regulated by the release of serotonin from gastrointestinal enterochromaffin cells.” should be changed into “The conclusion is that the regulation of depression induced by GEF-mediated alcohol withdrawal may be regulated by the release of 5-HT from EC cells.”.
- Line 300-301: “the Ginseng extract regulated depression is may be possible by the GEF found in ginseng.” should be changed into “It is possible that GEF of ginseng can regulate depression.”.
- Line 432-435: “The pharmacokinetics and pharmacodynamics studies need more work, to finely elucidate its mechanisms. More efforts are needed to develop of new methods to prepare pure Gintonin on an industrial scale, specifically for its use in long-term in Vivo and human studies, because of its composite nature of purification processes” should be “Pharmacokinetic and pharmacodynamic studies need to do more work, to better elucidate its mechanisms. More effort is needed to develop new methods to prepare pure gintonin on an industrial scale, specifically for long-term in vivo and human studies, Due to complex nature of its purification processes”.
- Line 444-451: “it needs a more clear understanding. The current findings have shown its beneficial effects against Alzheimer's disease, but the exact mechanisms are still not clear, for example, its effects against the Amyloid beta processing, whether it affects the enzymes responsible for the processing of the amyloid-beta? if yes, which specific enzyme, and its mechanism of action? All the related questions are still not answered. Similarly, it has been suggested that Gintonin is responsible for the protection of dopaminergic neurons, but still, the same questions are not answered? Almost all the studies have a very general concept, they haven’t given a detailed comprehensive touch to the mechanistic level.” Should be “and it needs clearer understanding. Current findings indicates its beneficial effects on AD, but the exact mechanisms are still not clear. For example, does its effects the enzymes responsible for the production of the Ab? If yes, which specific enzyme is involved in its mechanism of action? All the related questions are remain unanswered. Similarly, it has been suggested that gintonin may protect dopaminergic neurons, but the mechanism remain unanswered. Almost all the studies have only a very general concept, but the mechanism is not fully detailed.”
- Line453-456: “As noted in the studies conducted on Gintonin, in most cases the Gintonin is given orally, and the authors have concluded significant protective effects of Gintonin in different models of neurodegeneration, but at the same time, they haven’t analyzed the bioavailability of Gintonin. If its bioavailability is not significant, then how its availability may be improved?” Should be changed into “In most studies, gintonin was taken orally. Although gintonin has a significant protective effect in different neurodegenerative models, they have not yet analyzed the bioavailability of gintonin. If its bioavailability is not significant, then how to improve it?”.
- Line 461-462: “A pronounce amount of research is needed to explore this more safety, bioavailability and efficacy related parameters of gintonin.” should be changed into “A lot of research is needed to explore this gintonin parameter related to safety, bioavailability and efficacy.”.
- Line 471-473: “The qualities with their safety, efficacy and cheap availability make it the candidate drug for the management of neurodegenerative disorders to be further studied and presented for further pre-clinical and clinical studies.” Should be changed into “Before it becomes a drug candidate for the treatment of neurodegenerative diseases, its safety, effectiveness, and low-cost availability require further research and are provided to the next pre-clinical and clinical research.”.
- Line 108: the reference of Pyo, Mi-kyung, et al. (2011) is not cited in reference list.
- In Figure 2: the part of “Crude gintonin” should be put under “Bound compound (1 g)”; the part of “Gintonin-Enriched Fraction (GEF)” should be put under “Previpitate”.
- Figure 3 should be redraw to prevent the staggered of green line with red arrows.
- Figure 4 should be redraw to prevent the staggered of black lines with red arrows. The membrane bilayer pattern of Blood Brain Barrier should be replaced by cell layers.
- Figure 6: should be redraw. The chemical structure of ginsenoside shown in the figure is ginsenoside Rg3; “Increase in the intracellular Conc” should be “Increase in the intracellular Calcium level”; “Active mice” should be “Recovered mice”.
Minor Points:
- Line 21, 72, 140, 330, 332, 335, 338: “[Ca2C]i” and “[Ca2+]i” should be “[Ca2+]i”.
- Line 31: “neurodegenerative” should be “neurodegenerative diseases”.
- Line 37: “AD” should be “Alzheimer’s disease (AD)”.
- Line 41: “reactive oxygen” should be “reactive oxygen species (ROS)”.
- Line 42: “Normally oxygen” should be “Normally, oxygen”.
- Line 43: “moderately low, converts into” should be “moderately reduced, at low ratio, into”.
- Line 44, 316-317: “reactive oxygen species (ROS)” should be “ROS”.
- Line 60: “in Shi-Zhen Li in Ben Cao Gang Mu,” should be “in Ben Cao Gang Mu, written by Shi-Zhen Li”.
- Line 61: “Ming Reign in China” should be “Ming dynasty of China”.
- Line 62: “China (“dong bei“)” should be “northeast of China”.
- Line 64: ““cure-all“” should be ““cure-all“ in Greek”.
- Lin 67 and 70: “P. ginseng” should be in Italic.
- Line 71: “LPARs” should be “Lysophosphatidic acid (LPA) receptors”.
- Line 78: “Intraperitoneal” should be “intraperitoneal”.
- Lin 84: “Neurodegenerative” should be “neurodegenerative”.
- Line 108: “The gintonin was isolated by Pyo, Mi-kyung, et al. (2011)” should be “The gintonin was first isolated by Pyo, Mi-kyung, et al. (2011)”.
- Line 109: “methanol (CH3OH)” should be “methanol (MeOH)”.
- Line 113: “Ca2+-triggered Cl- channel” should be “Ca2+-triggered Cl- channel”.
- Line 114: “ethanol EtOH)” should be “ethanol (EtOH)”.
- Line 117: it is not clear for “to filter the mini part” mean.
- Line 119: reference 13 is not the right reference.
- Line 136: “Lysophosphatidic acid (LPA)” should be “LPA”.
- Line 139: reference 16 should be replaced by 20.
- Line 155-156: “Red arrows have shown …, Green lines are showing …. Red lines showing…” should be “Red arrows show…, Green lines show …. Red lines show…”.
- Line 168: reference 12 should be replaced by 21.
- Line 169: “PD is the most” should be “PD is the second most”.
- Line 174: “the expression nuclear factor erythroid 2-related factor 2 (Nrf2)” should be “the expression of Nrf2”.
- Line 178: “Here, we have” should be “In the report, we have”.
- Line 180: “DA” should be “dopaminergic”.
- Line 185-190: “MPTP(1-methyl-4-phenyl- 1,2,3,6-tetrahydropyridine), Nrf2; nuclear factor erythroid 2-related factor 2, … MPP+; 1-methyl-4-phenylpyridinium.” Should be “MPTP: 1-methyl-4-phenyl- 1,2,3,6-tetrahydropyridine), Nrf2: nuclear factor erythroid 2-related factor 2, …, MPP+: 1-methyl-4-phenylpyridinium.”.
- Line 194-195: “a danger to the health of elderly peoples around the globe.” Should be “a thread to the health of the elderly worldwide.”.
- Line 197: reference 23 is not the right reference.
- Line 202, 389: “Alzheimer’s disease” should be “AD”.
- Line 211: a reference of “Kim HJ, Shin EJ, Lee BH, Choi SH, Jung SW, Cho IH, Hwang SH, Kim JY, Han JS, Chung C, Jang CG, Rhim H, Kim HC, Nah SY. Oral Administration of Gintonin Attenuates Cholinergic Impairments by Scopolamine, Amyloid-β Protein, and Mouse Model of Alzheimer's Disease. Mol Cells. 2015 Sep;38(9):796-805.” should be added.
- Line 218: reference 20 should be added.
- Line 242: “GT” should be “gintonin”.
- Line 243: “3-NRA” should be “3-nitropropionic acid (3-NRP)”.
- Line 245: “AAV” should be “adeno-associated viral (AAV)”.
- Line 248: “CNS” should be “central nervous system (CNS)”.
- Line 254: “68e86” should be “68-86”.
- Line 258: “interferon-g, interleukin-6” should be “interferon-γ, IL-6”.
- Line 259: “insulin-like growth factorb1” should be “insulin-like growth factor-1”.
- Lin 260-261: “mitogen-activated protein kinase and nuclear factor-kB” should be “MAPK and NF-kB”.
- Line 265: “to explore about its mechanisms” should be “to explore its mechanisms”.
- Line 284: “GIT release 5-HT via Ca2+-reliant mechanism” should be “gastrointestinal tract (GIT) release of serotonin (5-HT) via calcium-reliant mechanism”.
- Line 286: “central nervous system” should be “CNS”.
- Line 293, 299: “serotonin” should be “5-HT”.
- Line 296-298: “The main concern over here is that, the possibility that gintonin-triggered release of catecholamine plasma, and acetylcholine and glutamate in the brains, may…” should be “The main concern is that the gintonin-triggered release of catecholamine, acetylcholine and glutamate in the brains may…”.
- Line 326: “[14, 15]” should be reformatted.
- Line 361: “LTP and LTD” should be “Long-Term Potentiation (LTP) and long-term depression (LTD)”.
- Line 395-398: “Sung Min Nam et all have…Gintonin against … that Gintonin significantly Improved the Hippocampal Long-Term Potentiation, improved Neurogenesis, and Cognitive…” should be “Nam et al. have …gintonin against … that gintonin significantly improved the hippocampal LTP, improved neurogenesis, and cognitive…”.
- Line 401-402: “choline acetyltransferase (ChAT) activity, and increased the level of acetylcholine esterase (AChE)” should be “ChAT activity, and increased the level of AChE”.
- Line 405: “Methylmercury-Induced Neurotoxicity” should be “methylmercury- induced neurotoxicity”.
- Line 407: “(Figure.8)” should be “(Figure 8)”.
- Line 413: “Nrf2; nuclear factor erythroid 2-related factor 2, HO-1; Heme…” should be “Nrf2: nuclear factor erythroid 2-related factor 2, HO-1: Heme…”.
- Line 428: “…aspects so that the claims are made more strong and effective” should be “…aspects to make the claims stronger and more effective”.
- Line 429: “protective in nature or restorative” should be “protective or restorative in nature”.
- Line 430: “have shown have given Gintonin co-treated with” should be “have shown that gintonin can be co-treated with”.
- Line 431-432: “be valuable for showing the mechanisms of action behind…” should be “be valuable to show the mechanisms behind…”.
- Line 441: “enzymes Gintonin biosynthesis” should be “enzymes in gintonin biosynthesis”.
- Line 451-452: “Gintonin may be considered as protective compounds” should be “gintonin become a protective compounds”.
- Line 457: “So, efforts can be made to make its stability and efficacy for the…” should be “Therefore, efforts may be made on its stability and efficacy in treating…”.
- Line 474: “be found as” should be “be developed as”.
Author Response
Reviewer#2
General Response: Thank you for reviewing our manuscript. The main points have been addressed accordingly, and have been presented comprehensively. We did our best to address all your comments and present our paper in a more clear and understandable manner. All the changes have been highlighted (Blue color) and respective line numbers have been given here in the response section. Please follow the provided Line numbers in the response sections, as the previously cited line numbers may have changed with the suggested correction/changes. We hope that this time the paper will be appropriate for publication in the journal of “cells”.
Reviewer#2
Major points:
Comment#1. In text, all “Gintonin” and “Glycogenolysis” should be presented as “gintonin” and “glycogenolysis”, respectively. “LPAR” and “LPAR1” should be “LPA receptor” and “LPA1 receptor”, respectively.
Response. All the indicated words have been corrected, which may be noted as highlighted with blue color. Page#11 and overall in the manuscript.
Comment#2. Line 274-276: “As, a current meta-analysis data has suggested that in depression, there is an elevation in the levels of different inflammatory-mediators including, tumor necrosis factor TNF-α, interleukin IL-6, and the IL-1 receptor antagonist compared to normal individuals” should be changed into “Therefore, the current meta-analysis data shows that compared with normal individuals, the levels of various inflammatory mediators (including tumor necrosis factor (TNF)-α, IL-6 and IL-1 receptor antagonists) in depression patients have increased.
Response. The sentence has been modified as suggested. Line number.270-272.
Comment#3. Line 296-298: “The main concern over here is that, the possibility that gintonin-triggered release of catecholamine plasma, and acetylcholine and glutamate in the brains, may alleviate the depressive-like effects.” should be changed into “The main concern is that the gintonin-induced release of catecholamine, acetylcholine and glutamate in the brain may alleviate depression-like effects.
Response. The sentence has been modified as suggested. Line number.292-296.
Comment#4. Line 298-300: “Conclusively, the GEF-mediated regulation of depression triggered by alcohol withdrawal may be regulated by the release of serotonin from gastrointestinal enterochromaffin cells.” should be changed into “The conclusion is that the regulation of depression induced by GEF-mediated alcohol withdrawal may be regulated by the release of 5-HT from EC cells.
Response. The sentence has been modified as suggested. Line number. 294-296.
Comment#5. Line 300-301: “the Ginseng extract regulated depression is may be possible by the GEF found in ginseng.” should be changed into “It is possible that GEF of ginseng can regulate depression.
Response. The sentence has been modified as suggested. Line number. 295-296.
Comment#6. Line 432-435: “The pharmacokinetics and pharmacodynamics studies need more work, to finely elucidate its mechanisms. More efforts are needed to develop of new methods to prepare pure Gintonin on an industrial scale, specifically for its use in long-term in Vivo and human studies, because of its composite nature of purification processes” should be “Pharmacokinetic and pharmacodynamics studies need to do more work, to better elucidate its mechanisms. More effort is needed to develop new methods to prepare pure gintonin on an industrial scale, specifically for long-term in vivo and human studies, Due to complex nature of its purification processes”.
Response. The sentence has been modified as suggested. Line number. 434-437.
Comment#7. Line 444-451: “it needs a more clear understanding. The current findings have shown its beneficial effects against Alzheimer's disease, but the exact mechanisms are still not clear, for example, its effects against the Amyloid beta processing, whether it affects the enzymes responsible for the processing of the amyloid-beta? if yes, which specific enzyme, and its mechanism of action? All the related questions are still not answered. Similarly, it has been suggested that Gintonin is responsible for the protection of dopaminergic neurons, but still, the same questions are not answered? Almost all the studies have a very general concept, they haven’t given a detailed comprehensive touch to the mechanistic level.” Should be “and it needs clearer understanding. Current findings indicates its beneficial effects on AD, but the exact mechanisms are still not clear. For example, does its effects the enzymes responsible for the production of the Ab? If yes, which specific enzyme is involved in its mechanism of action? All the related questions are remain unanswered. Similarly, it has been suggested that gintonin may protect dopaminergic neurons, but the mechanism remain unanswered. Almost all the studies have only a very general concept, but the mechanism is not fully detailed.”
Response. The sentence has been modified as suggested. Line number. 442-447.
Comment#8. Line453-456: “As noted in the studies conducted on Gintonin, in most cases the Gintonin is given orally, and the authors have concluded significant protective effects of Gintonin in different models of neurodegeneration, but at the same time, they haven’t analyzed the bioavailability of Gintonin. If its bioavailability is not significant, then how its availability may be improved?” Should be changed into “In most studies, gintonin was taken orally. Although gintonin has a significant protective effect in different neurodegenerative models, they have not yet analyzed the bioavailability of gintonin. If its bioavailability is not significant, then how to improve it?”.
Response. As suggested the sentence has been modified. Line number. 449-451.
Comment#9. Line 461-462: “A pronounce amount of research is needed to explore this more safety, bioavailability and efficacy related parameters of gintonin.” should be changed into “A lot of research is needed to explore this gintonin parameter related to safety, bioavailability and efficacy.
Response. The sentence has been modified as suggested. Line number. 456-459.
Comment#10. Line 471-473: “The qualities with their safety, efficacy and cheap availability make it the candidate drug for the management of neurodegenerative disorders to be further studied and presented for further pre-clinical and clinical studies.” Should be changed into “Before it becomes a drug candidate for the treatment of neurodegenerative diseases, its safety, effectiveness, and low-cost availability require further research and are provided to the next pre-clinical and clinical research.
Response. The sentence has been completely removed and modified completely. Line number. 454-458.
Comment#11. Line 108: the reference of Pyo, Mi-kyung, et al. (2011) is not cited in reference list.
Response. Thank you, it has been cited in the revised version. Line number. 107.
Comment#12. In Figure 2: the part of “Crude gintonin” should be put under “Bound compound (1 g)”; the part of “Gintonin-Enriched Fraction (GEF)” should be put under “Precipitate”.
Response. Yes, it has been modified as suggested. Figure.2
Comment#13. Figure 3 should be redraw to prevent the staggered of green line with red arrows. Done
Response. Yes, the figure has been redrawn and completely modified. It will look more simple and attractive. Figure.3.
Comment#14. Figure 4 should be redraw to prevent the staggered of black lines with red arrows. The membrane bilayer pattern of Blood Brain Barrier should be replaced by cell layers.
Response. The figure 4 has been completely revised and modified as suggested. Figure.4
Comment#15. Figure 6: should be redraw. The chemical structure of ginsenoside shown in the figure is ginsenoside Rg3; “Increase in the intracellular Conc” should be “Increase in the intracellular Calcium level”; “Active mice” should be “Recovered mice”.
Response. The figure.6 has been drawn again. Instead of ginsenoside Rg3 chemical structure, we have added just the roots of Ginseng.
Minor Points:
Comment#1. Line 21, 72, 140, 330, 332, 335, 338: “[Ca2C]i” and “[Ca2+]i” should be “[Ca2+]i”.
Response. The expression of “[Ca2+]i” has been changed to “[Ca2+]i”.
Comment#2. Line31: “neurodegenerative” should be “neurodegenerative diseases”.
Response. The “neurodegenerative” has been changed to “neurodegenerative diseases”. Line number. 35.
Comment#3. Line 37: “AD” should be “Alzheimer’s disease (AD)”.
Response. The “AD” has been changed to “Alzheimer’s disease (AD)”. Line number. 36.
Comment#4. Line 41: “reactive oxygen” should be “reactive oxygen species (ROS)”.
Response. The “reactive oxygen” has been changed to “reactive oxygen species (ROS)”. Line number. 37.
Comment#5. Line 42: “Normally oxygen” should be “Normally, oxygen”.
Response. Changed, as suggested. Line number. 38.
Comment#6. Line 43: “moderately low, converts into” should be “moderately reduced, at low ratio, into”. Line number. 39.
Response. The sentence moderately low, converts into” has been changed to “moderately reduced, at low ratio, into”. Line Number 38.
Comment#7. Line 44, 316-317: “reactive oxygen species (ROS)” should be “ROS”.
Response. The reactive oxygen species (ROS)” has been changed to “ROS”. Line Number 40.
Comment#8. Line 60: “in Shi-Zhen Li in Ben Cao Gang Mu,” should be “in Ben Cao Gang Mu, written by Shi-Zhen Li”.
Response. The sentence in Shi-Zhen Li in Ben Cao Gang Mu,” has been changed to “in Ben Cao Gang Mu, written by Shi-Zhen Li”. Line Number 56-57.
Comment#9. Line 61: “Ming Reign in China” should be “Ming dynasty of China”.
Response. The name “Ming Reign in China” has been changed to “Ming dynasty of China”. Line Number 57.
Comment#10. Line 62: “China (“dong bei“)” should be “northeast of China”.
Response. The “China (“dong bei“)” has revised and is written “northeast of China”. Line Number 58.
Comment#11. Line 64: ““cure-all“” should be ““cure-all“ in Greek”.
Response. The sentence cure-all“” has been written as ““cure-all“ in Greek. Line Number 60.
Comment#12. Line 67 and 70: “P. ginseng” should be in Italic.
Response. The names have been written italic. Line number. 63-66.
Comment#13. Line 71: “LPARs” should be “Lysophosphatidic acid (LPA) receptors”.
Response. The LPARs” has been presented as “Lysophosphatidic acid (LPA) receptors” throughout the manuscript. Line number. 135.
Comment#14. Line 78: “Intraperitoneal” should be “intraperitoneal”.
Response. Changed as pointed out. Line number. 74.
Comment#15. Lin 84: “Neurodegenerative” should be “neurodegenerative”.
Response. Changed as suggested. Line number. 80.
Comment#16. Line 108: “The gintonin was isolated by Pyo, Mi-kyung, et al. (2011)” should be “The gintonin was first isolated by Pyo, Mi-kyung, et al. (2011)”.
Response. Changed as indicated. Line number. 104.
Comment#17. Line 109: “methanol (CH3OH)” should be “methanol (MeOH)”.
Response. Done as suggested. Line number. 105.
Comment#18. Line 113: “Ca2+-triggered Cl- channel” should be “Ca2+-triggered Cl- channel”.
Response. The “Ca2+-triggered Cl- channel” has been written as “Ca2+-triggered Cl- channel”. Line number. 109.
Comment#19. Line 114: “ethanol EtOH)” should be “ethanol (EtOH)”.
Response. The “ethanol EtOH)” has been changed to “ethanol (EtOH)”. Line number. 109.
Comment#20. Line 117: it is not clear for “to filter the mini part” mean.
Response. The sentence has been modified as suggested. Line number.
Comment#21. Line 119: reference 13 is not the right reference.
Response. The reference has been revised as suggested.
Comment#22. Line 136: “Lysophosphatidic acid (LPA)” should be “LPA”.
Response. The Lysophosphatidic acid (LPA)” has been written as “LPA”. Line number. 135.
Comment#23. Line 139: reference 16 should be replaced by 20.
Response. Thank you, the reference 16 has been replaced by reference 20.
Comment#24. Line 155-156: “Red arrows have shown …, Green lines are showing …. Red lines showing…” should be “Red arrows show…, Green lines show …. Red lines show…”.
Response. Thank you, it has been revised as suggested. Line number. 152-153.
Comment#25. Line 168: reference 12 should be replaced by 21.
Response. The reference 12 has been replaced by 21.
Comment#26. Line 169: “PD is the most” should be “PD is the second most”.
Response. The “PD is the most” has been modified to “PD is the second most”. Line number. 164.
Comment#27. Line 174: “the expression nuclear factor erythroid 2-related factor 2 (Nrf2)” should be “the expression of Nrf2”.
Response. The nuclear factor erythroid 2-related factor 2 (Nrf2)” has been changed to “the expression of Nrf2”. Line number. 169
Comment#28. Line 178: “Here, we have” should be “In the report, we have”.
Response. The sentence “Here, we have” has been modified to “In the report, we have”. Line number. 173.
Comment#29. Line 180: “DA” should be “dopaminergic”.
Response. The “DA” has been written as “dopaminergic”. Line number. 175.
Comment#30. Line 185-190: “MPTP (1-methyl-4-phenyl- 1,2,3,6-tetrahydropyridine), Nrf2; nuclear factor erythroid 2-related factor 2, … MPP+; 1-methyl-4-phenylpyridinium.” Should be “MPTP: 1-methyl-4-phenyl- 1,2,3,6-tetrahydropyridine), Nrf2: nuclear factor erythroid 2-related factor 2, …, MPP+: 1-methyl-4-phenylpyridinium.”.
Response. The formatting has been rectified, as suggested. Line number.180-183.
Comment#31. Line 194-195: “a danger to the health of elderly peoples around the globe.” Should be “a thread to the health of the elderly worldwide.
Response. The sentence has been modified as suggested. Line number. 183.
Comment#32. Line 197: reference 23 is not the right reference.
Response. The reference has been removed, and rectified the overall sentence.
Comment#33. Line 202, 389: “Alzheimer’s disease” should be “AD”.
Response. The “Alzheimer’s disease” has been written as “AD. Line number. 194.
Comment#34. Line 211: a reference of “Kim HJ, Shin EJ, Lee BH, Choi SH, Jung SW, Cho IH, Hwang SH, Kim JY, Han JS, Chung C, Jang CG, Rhim H, Kim HC, Nah SY. Oral Administration of Gintonin Attenuates Cholinergic Impairments by Scopolamine, Amyloid-β Protein, and Mouse Model of Alzheimer's Disease. Mol Cells. 2015 Sep;38(9):796-805.” should be added.
Response. The reference has been added as suggested. Thank you.
Comment#35. Line 218: reference 20 should be added.
Response. The reference 20 has been added. Line number. 147.
Comment#36. Line 242: “GT” should be “gintonin”.
Response. The “GT” has been written as “gintonin”. Line number. 234.
Comment#37. Line 243: “3-NRA” should be “3-nitropropionic acid (3-NRP)”.
Response. The “3-NRA” has been changed to “3-nitropropionic acid (3-NRP)”. Line number. 235.
Comment#38. Line 245: “AAV” should be “adeno-associated viral (AAV)”.
Response. The AAV” has been written as “adeno-associated viral (AAV)”. Line number. 237.
Comment#39. Line 248: “CNS” should be “central nervous system (CNS)”.
Response. The CNS” has been presented as “central nervous system (CNS)”. Line number. 244.
Comment#40. Line 254: “68e86” should be “68-86”.
Response. The 68e86” has been corrected as “68-86. Line number. 248-249.
Comment#41. Line 258: “interferon-g, interleukin-6” should be “interferon-γ, IL-6”.
Response. The “interferon-g, interleukin-6” has been written as “interferon-γ, IL-6”. Line number. 251.
Comment#42. Line 259: “insulin-like growth factorb1” should be “insulin-like growth factor-1”.
Response. The sentence insulin-like growth factorb1” has rectifies as “insulin-like growth factor-1. Line number. 252
Comment#43. Lin 260-261: “mitogen-activated protein kinase and nuclear factor-kB” should be “MAPK and NF-kB”.
Response. Corrected as suggested. 256.
Comment#44. Line 265: “to explore about its mechanisms” should be “to explore its mechanisms”.
Response. Corrected as you have suggested. Line number. 258.
Comment#45. Line 284: “GIT release 5-HT via Ca2+-reliant mechanism” should be “gastrointestinal tract (GIT) release of serotonin (5-HT) via calcium-reliant mechanism”.
Response. The sentence has been revised as suggested. Line number. 277.
Comment#46. Line 286: “central nervous system” should be “CNS”.
Response. Corrected as suggested. Line number. 280.
Comment#47. Line 293, 299: “serotonin” should be “5-HT”.
Response. The “serotonin” has been changed to “5-HT”. Line number. 287.
Comment#48. Line 296-298: “The main concern over here is that, the possibility that gintonin-triggered release of catecholamine plasma, and acetylcholine and glutamate in the brains, may…” should be “The main concern is that the gintonin-triggered release of catecholamine, acetylcholine and glutamate in the brains may…”.
Response. The sentence has been revised as suggested, which may be read at: Line number. 290-293.
Comment#49. Line 326: “[14, 15]” should be reformatted.
Response. The sentences have been completely modified as suggested.
Comment#50. Line 361: “LTP and LTD” should be “Long-Term Potentiation (LTP) and long-term depression (LTD)”.
Response. The “LTP and LTD” have been revised to “Long-Term Potentiation (LTP) and long-term depression (LTD)”. Line number. 354.
Line number.
Comment#51. Line 395-398: “Sung Min Nam et all have…Gintonin against … that Gintonin significantly Improved the Hippocampal Long-Term Potentiation, improved Neurogenesis, and Cognitive…” should be “Nam et al. have …gintonin against … that gintonin significantly improved the hippocampal LTP, improved neurogenesis, and cognitive…”.
Response. The sentence has been revised as suggested. Line number 389.
Comment#52. Line 401-402: “choline acetyltransferase (ChAT) activity, and increased the level of acetylcholine esterase (AChE)” should be “ChAT activity, and increased the level of AChE”.
Response. Revised as suggested. Line number. 394-395.
Comment#53. Line 405: “Methylmercury-Induced Neurotoxicity” should be “methylmercury- induced neurotoxicity”.
Response. Revised as suggested. Line number. 398.
Comment#54. Line 407: “(Figure.8)” should be “(Figure 8)”.
Response. Corrected as suggested. Line number.400.
Comment#55. Line 413: “Nrf2; nuclear factor erythroid 2-related factor 2, HO-1; Heme…” should be “Nrf2: nuclear factor erythroid 2-related factor 2, HO-1: Heme…”.
Response. The formatting has been done, as suggested. Line number. 405-408.
Comment#56. Line 428: “…aspects so that the claims are made more strong and effective” should be “…aspects to make the claims stronger and more effective”.
Response. The sentence has been revised as suggested. Line number. 427-428.
Comment#57. Line 429: “protective in nature or restorative” should be “protective or restorative in nature”.
Response. The sentence has been revised as suggested. Line number. 429.
Comment#58. Line 430: “have shown have given Gintonin co-treated with” should be “have shown that gintonin can be co-treated with”.
Response. The sentence has been modified as suggested. Line number. 430.
Comment#59. Line 431-432: “be valuable for showing the mechanisms of action behind…” should be “be valuable to show the mechanisms behind…”.
Response. The sentence has been revised as suggested. Line number. 431-432.
Comment#60. Line 441: “enzymes Gintonin biosynthesis” should be “enzymes in gintonin biosynthesis”.
Response. The current sentence has been extensively revised, we have that the current wording will be acceptable and easily understandable.
Comment#61. Line 451-452: “Gintonin may be considered as protective compounds” should be “gintonin become a protective compounds”.
Response. The sentence has been revised as suggested. Line number. 445.
Comment#62. Line 457: “So, efforts can be made to make its stability and efficacy for the…” should be “Therefore, efforts may be made on its stability and efficacy in treating…”.
Response. The overall sentence has been revised and rearranged.
Comment#63. Line 474: “be found as” should be “be developed as”.
Response. The complete sentence has been revised and edited as suggested by the other reviewer too. Thank you for your time and a deep reading and reviewing our manuscript.
Thank you so much and best regards for your time and highly constructive comments and suggestions.
Kind Regards:

Round 2
Reviewer 2 Report
I believe the manuscript has been significantly improved and now warrants publication in Cells.